# Effect of Fluid Intake on Acute Changes in Plasma Volume: A Randomized Controlled Crossover Pilot Trial

**DOI:** 10.3390/metabo14050263

**Published:** 2024-05-06

**Authors:** Janis Schierbauer, Sabrina Sanfilippo, Auguste Grothoff, Ulrich Fehr, Nadine Wachsmuth, Thomas Voit, Paul Zimmermann, Othmar Moser

**Affiliations:** 1Division of Exercise Physiology and Metabolism, Bayreuth Centre of Sport Science, University of Bayreuth, 95440 Bayreuth, Germany; sabrina.sanfilippo@uni-bayreuth.de (S.S.); auguste.grothoff@uni-bayreuth.de (A.G.); ulrich.fehr@uni-bayreuth.de (U.F.); nadine.wachsmuth@uni-bayreuth.de (N.W.); thomas.voit@uni-bayreuth.de (T.V.); paul.zimmermann@uni-bayreuth.de (P.Z.); othmar.moser@uni-bayreuth.de (O.M.); 2Interdisciplinary Metabolic Medicine Trials Unit, Department of Internal Medicine, Division of Endocrinology and Diabetology, Medical University of Graz, 8036 Graz, Austria

**Keywords:** blood volume, total body water, hemoglobin concentration, hematocrit, sodium-chloride, Ringer, hydration

## Abstract

Plasma volume (PV) undergoes constant and dynamic changes, leading to a large intra-day variability in healthy individuals. Hydration is known to induce PV changes; however, the response to the intake of osmotically different fluids is still not fully understood. In a randomized controlled crossover trial, 18 healthy individuals (10 females) orally received an individual amount of an isotonic sodium-chloride (ISO), Ringer (RIN), or glucose (GLU) solution. Hemoglobin mass (Hbmass) was determined with the optimized carbon monoxide re-breathing method. Fluid-induced changes in PV were subsequently calculated based on capillary hemoglobin concentration ([Hb]) and hematocrit (Hct) before and then every 10 minutes until 120 min (t_0–120_) after the fluid intake and compared to a control trial arm (CON), where no fluid was administered. Within GLU and CON trial arms, no statistically significant differences from baseline until t_120_ were found (*p* > 0.05). In the ISO trial arm, PV was significantly increased at t_70_ (+138 mL, *p* = 0.01), t_80_ (+191 mL, *p* < 0.01), and t_110_ (+182 mL, *p* = 0.01) when compared to t_0_. Moreover, PV in the ISO trial arm was significantly higher at t_70_ (*p* = 0.02), t_110_ (*p* = 0.04), and t_120_ (*p* = 0.01) when compared to the same time points in the CON trial arm. Within the RIN trial arm, PV was significantly higher between t_70_ and t_90_ (+183 mL, *p* = 0.01) and between t_110_ (+194 mL, *p* = 0.03) and t_120_ (+186 mL, *p* < 0.01) when compared to t_0_. These results demonstrated that fluids with a higher content of osmotically active particles lead to acute hemodilution, which is associated with a decrease in [Hb] and Hct. These findings underpin the importance of the hydration state on PV and especially on PV constituent levels in healthy individuals.

## 1. Introduction

The blood is a vital organ and complex emulsion within the human body. Total blood volume (BV) makes up about 7–8% of an adult’s body mass and comprises various cellular components and a liquid extracellular matrix, i.e., the plasma volume (PV) [1]. The ratio between the cellular and liquid components is expressed via the hematocrit (Hct). The BV plays a central role in maintaining homeostasis, transporting nutrients, hormones, and waste products, as well as supporting the immune system. In the context of exercise physiology, the BV is a major limiting factor of endurance exercise performance, directly influencing maximum oxygen uptake via its influence on cardiac output and arterio-venous oxygen difference [2,3,4,5].

It has been demonstrated that the cellular components of BV, especially red cell mass, are stable over short (and long) periods of time in healthy individuals [6]. In contrast, the PV is not static; rather, it undergoes constant and dynamic changes leading to a large intra-day variability [7,8]. It is therefore influenced by various factors, e.g., the hydration state, heat or altitude exposure, and physical exercise. During heat and altitude exposure, for instance, PV is known to acutely decrease and these physiological changes are associated with hemoconcentration leading to an improved tissue oxygen supply at high altitudes [9]. In the context of physical exercise, PV is known to follow three opposing and time-dependent trends. Firstly, PV is acutely reduced, which was shown for both endurance and resistance exercise [2,3,10]; secondly, it is restored within the first two hours post-exercise; and thirdly, it is significantly elevated within 24-h post-exercise, given adequate hydration [11]. Moreover, after repeated and systematic exercise training, PV is further and constantly increased [12].

This physiological regulation makes it clear that hematological concentrations, e.g., [Hb] and Hct are directly linked to PV, which is why they tend to demonstrate a large within- and between-subject variability. Nonetheless, Hct and [Hb] are very useful in calculating the exact fluctuations of PV for both acute and chronic changes, provided total hemoglobin mass (Hbmass) is known, e.g., after carbon-monoxide rebreathing [13]. In this context, one of the most important factors that can acutely influence PV is the hydration state [14]. This is primarily affected by the amount of fluid that is absorbed and excreted by the body. Both processes have been shown to have a great influence on total PV but the exact extent depends on the type of fluid being consumed, particularly its content of osmotically active particles. To date, it remains unclear what influence different osmotic agents have on acute changes in the PV under resting conditions.

Therefore, the aim of this single-center randomized controlled crossover pilot trial was to investigate the acute effect of different osmotically active fluids on the PV response at rest in moderately active adults with no cardiac abnormalities.

## 2. Materials and Methods

This single-center randomized controlled crossover trial is a secondary outcome analysis of a previously published trial [15] and investigated the effects of different orally administered fluids on changes in PV. The local ethics committee of the University of Bayreuth (Germany) approved the study protocol (O 1305/1—GB, 10 June 2021). The study was conducted in conformity with the declaration of Helsinki and guidelines for Good Clinical Practice. Before any trial-related activities, potential participants were informed about the study protocol and participants gave their written informed consent.

### 2.1. Eligibility Criteria

Eligibility criteria included male or female individuals aged 18–35 years with a body mass index (BMI) of 18.0–29.9 kg·m^−2^ (both inclusive). Individuals were excluded if they were enrolled in a different study, received any pharmaceutical products including investigational medications, or had blood pressure outside of the range of 90–150 mmHg for systolic and 50–95 mmHg for diastolic after resting for five minutes in a supine position. Furthermore, participants were excluded if they suffered from any metabolic disease, including renal or thyroid, or had a history of multiple and/or severe allergies to any trial-related products. In addition, females who were pregnant or individuals with a heart pacemaker were also excluded. To assure a euhydrated status prior to the study experiments, participants were also excluded if they demonstrated a urine-specific gravity outside the range of 1005–1030 mg·mL^−1^.

### 2.2. Assessment of Eligibility

Inclusion and exclusion criteria were assessed by an investigator at the screening visit prior to the start of the study.

### 2.3. Study Design

This study consisted of six study-specific visits, including a screening visit (V1) and five trial-related visits (V2–6). After inclusion in the study, participants were assigned to ascending numbers and then allocated to the order in which the trial visits were conducted following a crossover randomized fashion (1:1:1:1) with the software Research Randomizer^®^ (Version 4.0) [16]. Based on their body mass, which was measured at the start of each trial visit, participants received an individual quantity of each solution. Hemoglobin concentration ([Hb]) and hematocrit (Hct) were analyzed before, immediately after, and then every 10 min for 120 min after the fluid consumption using capillary blood samples from the earlobe. Within three days after V5, an optimized carbon-monoxide re-breathing method was conducted to determine the hemoglobin mass (Hbmass) and blood volume. Between each trial visit, a minimum of 48 h was maintained, except for the control condition, after which the next visit could take place after a minimum of 24 h.

### 2.4. Study Visits

#### 2.4.1. Screening (V1) and Fluid Consumption (V2–5)

At a screening visit, participants were assessed by the same investigator for inclusion and exclusion criteria, assigned to a study ID, and informed about the randomization of the following four study visits. Prior to the start of each of these trial visits, participants had to fast for at least 12 h and refrain from any strenuous physical activity for at least 48 h. Participants were also not allowed to consume alcohol within 24 h before each visit. Upon arrival at the research facility of the University of Bayreuth in the morning after an overnight fast, participants were asked to use the bathroom for bowel emptying, if necessary. A urine-specific gravity test was performed to ensure a euhydrated status (Combur10, Roche Deutschland Holding GmbH, Grenzach-Whylen, Germany). The total body water (TBW) including intra- and extracellular water, skeletal muscle, and fat mass was assessed using a bio-electrical impedance analysis (InBody 720, InBody Co., Seoul, South Korea). The participants then received 12 mL per kg body mass of either isotonic sodium chloride (NaCl 0.9%, B. Braun, B. Braun Melsungen AG, Melsungen, Germany), 5% glucose (Glucose-Lösung 5%, Deltamedica, Reutlingen, Germany), or Ringer’s solution (Ringer B. Braun, Melsungen AG, Melsungen, Germany), which had to be consumed within 60 s (for electrolyte composition and osmolarity of the different solutions see Table 1). During the control visit, no liquids were consumed. Capillary blood samples from the earlobe for the measurement of hemoglobin concentration ([Hb]) and hematocrit (Hct) were obtained before the fluid consumption (t_0_) and then every 10 min for 120 min after the fluid consumption (t_10–120_). [Hb] was analyzed immediately with a point-of-care hemoglobin analyzer (Hemocue Hemoglobin 201, Radiometer, Krefeld, Germany), while Hct was first centrifuged and then read out manually via micro-hematocrit. Any other form of food and fluid intake was not allowed during each of the trial visits. The trial arms are subsequently abbreviated as CON (control), GLU (glucose), ISO (isotonic sodium-chloride), and RIN (Ringer). Throughout every study visit, participants were kept under standardized room temperature (22 °C) and humidity (50%).

#### 2.4.2. Optimized Carbon-Monoxide Rebreathing Method (V6)

The total hemoglobin mass (Hbmass), blood volume (BV), plasma volume (PV), and erythrocyte volume (ECV) were determined using a carbon monoxide (CO)-rebreathing procedure according to methods described in previous investigations [13,17,18]. In brief, an individual dose of CO (0.8–0.9 mL·kg^−1^, CO 3.7, Linde AG, Unterschleißheim, Germany) was administered and rebreathed along with 3 L of pure medical oxygen (Med. O_2_ UN 1072, Rießner-Gase GmbH, Lichtenfels, Germany) for 2 min. Capillary blood samples were taken before and 6- and 8-min post administration of the CO dose. The blood samples were measured for the determination of %HbCO using an OSM III hemoximeter (Radiometer, Copenhagen, Denmark). The Hbmass was then calculated based on the mean percentage change in %HbCO before and after the CO was rebreathed. For the calculation of PV changes, all capillary [Hb] were converted to the venous conditions [19,20]. The BV, ECV, and PV were calculated according to the following formulas, where 0.91 = cell factor at sea level [21]:$$BV\left(mL\right)=Hbmass\left(g\right)\times 100÷[Hb]{(g\cdot dL}^{-1})÷0.91$$
$$ECV\left(mL\right)=Hbmass\left(g\right)÷\left(\left[Hb\right]\left({g\cdot dL}^{-1}\right)÷Hct\times 100\right)\times 100$$
$$PV\left(mL\right)=BV\left(mL\right)-ECV(mL)$$

Since the Hbmass does not change over short periods [22], the temporary offset determination of the [Hb] for the calculation of the BV is possible without compromising accuracy. For a detailed description of the accuracy of the methods, see [13,17,18]. The typical error for Hbmass in our laboratory is 1.5%, which is comparable to previous investigations [19,23], while the typical error for BV is 2.5%.

### 2.5. Statistical Analysis

All data were assessed for normal distribution by means of the Shapiro–Wilk normality test. Descriptive statistics are given as mean ± standard deviation (SD), interquartile range (IQR) and minima and maxima. To analyze for within- and between-trial arm changes, a two-way repeated measures ANOVA or a mixed-effects model with Geisser-Greenhouse correction followed by post-hoc Tukey multiple comparisons test were performed. Statistical analyses were conducted using GraphPad Prism Software version 8.0 (GraphPad, La Jolla, CA, USA).

## 3. Results

In total, 18 healthy individuals (10 females) were included for statistical analyses with a mean ± SD age of 23.1 ± 1.8 years, height of 176 ± 10 cm, body mass of 69.5 ± 12.5 kg, and BMI of 22.2 ± 2.1 kg·m^−2^. All screened participants were eligible to be part of the study, in which no participant had to be excluded or left the study prematurely.

Based on the individual body mass on the days of the fluid conditions, participants consumed 831.6 ± 148.9 mL of ISO, 835.3 ± 151.3 mL of GLU, and 832.7 ± 149.4 mL of RIN (*p* > 0.99), equaling a ~2% increase in total body water.

At the day of the CO-rebreathing, Hbmass and BV were 797 ± 227 g (10.7 ± 3.2 g·kg^−1^) and 5920 ± 1321 mL (86 ± 12 mL·kg^−1^), respectively. PV and ECV were 3618 ± 746 mL (52.8 ± 8.3 mL·kg^−1^) and 2302 ± 603 mL (47.5 ± 8.6 mL·kg^−1^), respectively. Baseline values of [Hb], Hct, and PV at t_0_ were not statistically different between trial arms (all *p* > 0.98).

Figure 1 presents the absolute changes in [Hb] and Hct over the course of the 120-min observational phase. A two-way repeated measures ANOVA revealed that there was no statistically significant time or trial arm effect on [Hb] and Hct in both the CON (*p* = 0.96–0.99), GLU (*p* = 0.75–0.99), and RIN (*p* = 0.52–0.99) trial arms. In the ISO trial arm, [Hb] was significantly lower at t_70_ (14.3 ± 1.4 vs. 13.9 ± 1.4 g·dL^−1^, *p* = 0.02) and t_80_ (13.8 ± 1.3 g·dL^−1^, *p* < 0.01) when compared to t_0_. All other within-trial arm comparisons were insignificant (*p* = 0.19–0.99).

Hct was significantly lower, by 1.8% at t_110_ (*p* < 0.01) and 1.7% at t_120_ (*p* = 0.01) in the RIN trial arm. All other within-trial arm comparisons were insignificant (*p* = 0.08–0.99). In the ISO trial arm, Hct was 1.4% lower at t_110_ when compared to baseline; however, this difference was not statistically significant (*p* = 0.08). All other within-trial arm comparisons were also insignificant (*p* = 0.12–0.99). In the CON (all *p* > 0.99) and GLU trial arms (*p* = 0.98–0.99), no statistically significant differences from the baseline were found (see Figure 1).

With regard to changes in PV, no statistically significant differences from baseline until t_120_ were found in the CON (*p* = 0.59–0.99) and GLU trial arms (*p* = 0.97–0.99, see Figure 2). In the ISO trial arm, PV was significantly increased at t_70_ (+138 mL, *p* = 0.01) and t_80_ (+191 mL, *p* < 0.01) when compared to t_0_. The difference between t_0_ and t_110_ was also significant (*p* = 0.01). All other within-trial arm comparisons were insignificant (*p* = 0.13–0.99). Moreover, PV in the ISO trial arm was significantly higher at t_70_ (*p* = 0.02), t_110_ (*p* = 0.04), and t_120_ (*p* = 0.01) when compared to the same time points in the CON trial arm.

In the RIN trial arm, PV was significantly higher between t_70_ and t_90_ (+183 mL, *p* = 0.01) and between t_110_ (+194 mL, *p* = 0.03) and t_120_ (+186 mL, *p* < 0.01) when compared to t_0_. All other within-trial arm comparisons were insignificant (*p* = 0.15–0.99).

## 4. Discussion

This study aimed to investigate the effects of various osmotically different liquid solutions on acute changes in PV in healthy individuals. Changes in PV were calculated until 120 min post-fluid consumption based on capillary [Hb] and Hct. Our results demonstrated that the electrolyte composition and osmolarity of the Ringer and isotonic sodium-chloride solutions induce a significant increase in total PV. In contrast, PV remained unchanged in the CON and GLU trial arms.

This is the first trial that investigated absolute PV changes under both resting conditions and after the consumption of osmotically different solutions. Our findings demonstrate that the ISO and RIN solutions induced a significant 5% and 5.3% increase in PV, respectively. These findings are only marginally lower than the 7.5% increase that was found after glycerol hyperhydration under resting conditions [14]. This state, referred to as hypervolemia, is the result of an increased osmotic load, which in turn increases the medullary concentration gradient in the kidneys, leading to water reabsorption. This physiologic regulation would also explain why the GLU group showed no significant changes in PV compared to the baseline. The glucose within the solution is rapidly taken up by the cells, therefore not contributing to an increased osmotic load [24]. The latter is also reflected in the generally lower osmotically active particle content in the 5% glucose solution compared to isotonic sodium-chloride and Ringer’s solution (see Table 1).

These dynamic PV changes are generally well known; however, their quantitative extent is still insufficiently understood. For a healthy population, it has been repeatedly demonstrated that PV is associated with large intra-day variability, which is mostly due to posture effects, hydration, sleep, or food intake [7,24]. This can even be further enhanced in various disease states, e.g., chronic hypertension [25,26], chronic kidney disease [27], or chronic fatigue syndrome [28].

In addition to hydration, which was the prime mediator in this study, PV in healthy individuals is also markedly affected by heat or altitude exposure as well as physical exercise. The latter is of special importance as PV during exercise is substantially decreased while in the long-term, e.g., after systematic endurance training, it is significantly increased [12,29]. This is why it was previously suggested that conclusions based on changes in levels of plasma constituents without evaluation of associated changes in PV can lead to erroneous interpretations [30,31]. For this reason, it was recommended to consider the effect of PV changes on blood constituent levels in order to allow for accurate and informed intra- and inter-individual comparisons. However, most studies still neglect this effect presenting uncorrected values. In this study, plasma constituents, i.e., [Hb] and Hct, were markedly affected by the RIN and ISO solutions, which were accompanied by hemodilution. From a practical perspective, these findings are of special importance for blood donations, where suitability for a donation (or a repeated donation) is determined via [Hb].

The fact that the PV in the ISO or RIN trial arms was not consistently higher than the CON trial arm probably stems from the minor fluctuations in [Hb] and Hct at the specific time points, respectively (see Figure 1). Since we calculated PV changes based on capillary [Hb] and Hct, sampling errors could lead to inaccurate measurements, although we performed repeated punctures of the hyperemized earlobe before taking our capillary samples. Moreover, this was a secondary outcome analysis of a previously published study where we investigated the effects of these osmotically different solutions on changes in body composition. Here, body composition was assessed every 10 min using a bio-electrical impedance analysis. This procedure required the participants to transfer from a seated into a standing position which was then maintained for the duration of the analysis (1 min) before they transferred back to a seated position. Measurements of [Hb] and Hct were taken in advance of the posture change in order to diminish the effects of posture on PV and thus [Hb] and Hct. Since the participants then remained in a seated position for 10 min before the next capillary measurement, any posture-associated PV changes would have been diminished by then. We can therefore safely assume that in this study, no effect of posture on the observed changes in PV exists. This is also supported by the fact that we found no changes in PV in the CON group and that sitting in general was reported to have only a little effect on PV changes.

Our findings also provide some practical implications. It is well known that a euhydrated state is crucial for achieving and maintaining peak performance. For instance, during exercise, achieving a high cardiac output and thus maximum oxygen uptake is directly dependent on the circulating BV [32]. The higher the BV, the higher the venous return to the pumping heart and thus the end-diastolic volume, which results in a larger stroke volume. A higher stroke volume is not only seen in athletes with a larger BV but also in untrained subjects after an intravenous volume infusion [33]. Given the fact that during exercise BV tends to substantially decrease and thus impair stroke volume and oxygen uptake, a euhydrated or even hyperhydrated state could mitigate the exercise-induced volume shifts to the extent that an impairment of the stroke volume response would be reduced. It must be stated, however, that any hypervolemia would also lead to a decrease in Hct and [Hb] and thus arterial oxygen content. The latter is especially associated with a decrease in oxygen transport capacity, which can be calculated if oxygen saturation levels and cardiac output are also known. On the one hand, it is likely that fluid-induced hypervolemia would increase cardiac output but simultaneously decrease arterial oxygen content. In this context, we were recently able to demonstrate that during aerobic and anaerobic exercise, the exercise-induced PV shifts are very much associated with a decrease in stroke volume and thus cardiac output; however, due to the hemoconcentration, the arterial oxygen content was significantly increased leading to an overall increase in oxygen transport capacity [2,3]. To date, it remains unknown how the opposing adaptations (hypervolemia) would ultimately affect oxygen transport capacity.

Of course, this study is not without limitations. First, the results obtained from this study come from healthy young adult participants. Moreover, all of them were sports economics students who engage in regular physical exercise both during their studies and in their everyday lives. Although we did not use evaluation measures to assess the types of intensity of physical activity and sitting time that our participants do as part of their daily lives (e.g., to estimate total physical activity and sitting time), we would argue that this population is best described as above average active, which is also confirmed by the values for Hbmass and BV found in this study when compared to reference values [34]. It remains unknown whether the same extent of hemodilution would appear in sedentary or sick individuals. Lastly, it was previously reported that the menstrual cycle-induced hormone changes can have profound effects on total body water (and thus hemoconcentrations) in women [35,36]. As we did not control for the course of the menstrual cycle in our female participants, we cannot tell whether the measured fluctuations are (in part) due to a specific phase of their menstrual cycle. Future studies should therefore incorporate the control of the phases of the menstrual cycle when recruiting females participants.

## 5. Conclusions

Orally administered Ringer and isotonic sodium-chloride solutions significantly increase total PV in healthy moderately-trained individuals. This was accompanied by a concomitant decrease in [Hb] and Hct due to hemodilution. At the same time, no changes in PV were found after the intake of a 5% glucose solution. PV also remained unchanged under control conditions, making the acute PV changes found in the RIN and ISO trial arms due to the osmolarity of these solutions. Our results demonstrated the impact of osmotically different liquid solutions on acute changes in PV offer new insights in the balancing of an adequate hydration state in healthy individuals.

## Figures and Tables

**Figure 1 metabolites-14-00263-f001:**
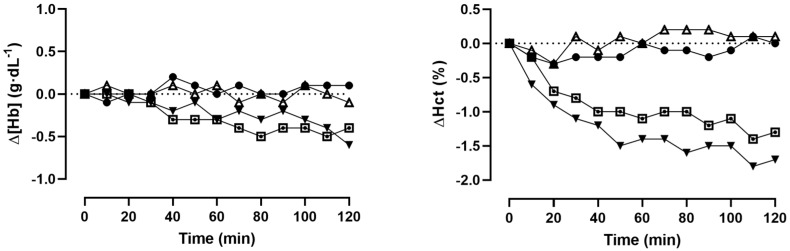
Absolute changes in [Hb] and Hct from baseline (t_0_) until 120 min (t_120_) post fluid consumption (● = CON, ▲ = GLU, ▼ = RIN, ⊡ = ISO).

**Figure 2 metabolites-14-00263-f002:**
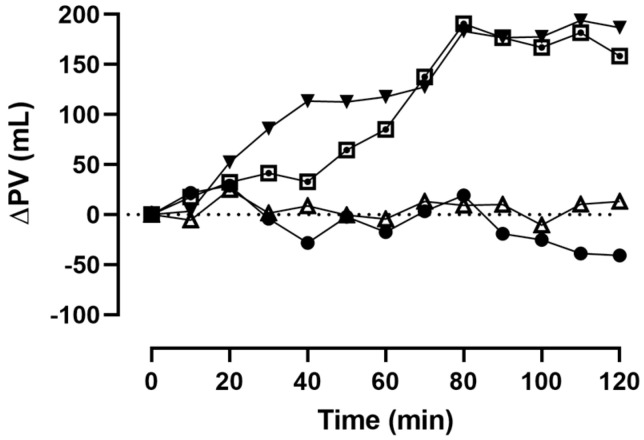
Absolute changes in PV from the baseline (t_0_) until 120 min (t_120_) post fluid consumption (● = CON, ▲ = GLU, ▼ = RIN, and ⊡ = ISO).

**Table 1 metabolites-14-00263-t001:** Electrolyte composition and osmolarity of the sodium-chloride, Ringer, and 5% glucose (G5) solution.

	Sodium-Chloride	Ringer	5% Glucose
Sodium (mmol∙L^−1^)	154	147	0
Chloride (mmol∙L^−1^)	154	156	0
Potassium (mmol∙L^−1^)	0	4	0
Calcium (mmol∙L^−1^)	0	2.2	0
Glucose (g∙L^−1^)	0	0	50
Osmotically active particles (mOsm∙L^−1^)	308	309	278

## Data Availability

Data will be made available upon reasonable request by the corresponding author.

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
