# Peer review of "Effect of Fluid Intake on Acute Changes in Plasma Volume: A Randomized Controlled Crossover Pilot Trial"

_metabolites, 2024, doi:10.3390/metabo14050263_

Round 1
Reviewer 1 Report
Comments and Suggestions for Authors
It is stated that the participants of the study consisted of moderately active individuals. There is no assessment method for determining the activity levels of individuals. It was decided according to what they were moderately active. This issue can perhaps be considered as a limitation of the study.
What is G5 in Table 1, which is understood to refer to the glucose group, should be specified in the table. Because glucose is abbreviated as GLU in the text.
The limitations of the study were not stated.
Author Response
Dear Reviewer,
thank you very much for your time and effort in reviewing our manuscript. We have found your comments to be very helpful in optimizing the general qualitity of the submission. Please find here our detailed replys to your comments:
It is stated that the participants of the study consisted of moderately active individuals. There is no assessment method for determining the activity levels of individuals. It was decided according to what they were moderately active. This issue can perhaps be considered as a limitation of the study.
Answer: "We agree. Usually we use data from the IPAQ-SF to determine the activity levels of our participants, however, we did not in this study. So we would suggest to write healthy individuals. This would be accurate without the need to provide an assessment method. For your information, we also discussed this part in the limitations section (see below). Please feel free to give us your opinion on that. Thank you."
What is G5 in Table 1, which is understood to refer to the glucose group, should be specified in the table. Because glucose is abbreviated as GLU in the text.
Answer: "We agree and have revised both the tabel and the table heading."
The limitations of the study were not stated.
Answer: "We have added an limitations section at the end of the discussion. We hope this improves the quality of this section."
If you have any further questions, do not hesitate to contact me.
Best wishes
Dr. Janis Schierbauer
Reviewer 2 Report
Comments and Suggestions for Authors
The paper is generally well-written and the methods are thoroughly described. The results section can be improved by clearly reporting the omnibus test results reflecting significant interactions where applicable. As currently presented, it’s unclear why some post-hoc comparisons are reported while others are not.
Comments on the Quality of English LanguageLine 14 – suggest removal of “Especially”
Line 58 – reconsider the use of “fundament” here
Line 58 – missing word in “tend to a”?
Line 176 – not sure if TBW was defined previously and it’s unclear if the reported numbers reflect body mass or total body water values from BIA?
Line 270 – assuming SV is stroke volume, the abbreviation is unnecessary here
Author Response
Dear Reviewer,
thank you very much for your time and effort in reviewing our manuscript. We have found your comments to be very helpful in optimizing the general qualitity of the submission. Please find here our detailed replys to your comments:
As currently presented, it’s unclear why some post-hoc comparisons are reported while others are not.
Answer: "Thank you for your comment. We agree with your statement and have added the necessary post-hoc values in the results section."
Line 14 – suggest removal of “Especially”
Answer: "Thank you, we have revised."
Line 58 – reconsider the use of “fundament” here
Answer: "Thank you, we agree and have revised."
Line 58 – missing word in “tend to a”?
Answer: "Thank you, we have revised."
Line 176 – not sure if TBW was defined previously and it’s unclear if the reported numbers reflect body mass or total body water values from BIA?
Answer: "Thank you, we have revised. We have also added further information in the methods section in regard to the BIA and the parameters we obtained (L. 119-121)."
Line 270 – assuming SV is stroke volume, the abbreviation is unnecessary here
Answer: "We agree and have revised."
If you have any further questions, do not hesitate to contact me.
Best wishes
Dr. Janis Schierbauer